# Inappropriate timing of salt intake increases the risk of heat-related illness: An observational study

**Takeyasu Kakamu** [1]*, **Teruna Ito**[2], **Shota Endo**[1], **Tomoo Hidaka**[1], **Yusuke Masuishi**[1], **Hideaki Kasuga**[1], **Tetsuhito Fukushima**[1]

**1** Department of Hygiene and Preventive Medicine, Fukushima Medical University, Fukushima, Fukushima, Japan, **2** Department of Food and Nutrition, Koriyama Women's University, Koriyama, Fukushima, Japan

* bamboo@fmu.ac.jp

## Abstract

The importance of salt intake in preventing heat-related illness (HRI) is well established, however, the specific method of ingestion has not been sufficiently studied. This study, therefore, aimed to investigate the optimal timing of salt intake to prevent HRI during hot outdoor work. We recruited 28 healthy male firefighters working at a fire department in Japan. They were provided a questionnaire to complete before and after receiving training in the summer season. We assessed their salt intake as before, during, and after training or none. In addition, they completed a brief self-administered diet history questionnaire to evaluate their daily salt and alcohol intake. HRI was determined through subjective and objective symptoms listed in the questionnaire, and environmental data were obtained from a national database. Subsequently, factors related to HRI were determined using a logistic regression model. The mean age of the participants was 31.0 ± 7.7 years. The study was performed within 250 working days, and we detected 28 HRI symptoms (11.2%). The median alcohol intake was 25.6 g/day when calculated according to the actual work system. Logistic mixed effect model analysis revealed that salt intake before training (OR: 5.893, 95% CI: 1.407–24.675), and salt intake before and during training (OR: 22.889, 95% CI: 4.276–122.516) were positively associated with HRI symptoms. The results indicate that inappropriate timing of salt intake increases the risks of HRI. Thus, a timely intake of salt in adequate amounts may be important in preventing these risks.

## Introduction

Heat-related illness (HRI) is a common occupational injury that must be addressed. Heat-related morbidity and mortality are expected to escalate as climate change continues due to global warming, with each additional unit rise in temperature projected to further increase these risks [1]. Most outdoor workers are repeatedly exposed to daily occupational heat stress over extended periods, making them more susceptible to both acute and chronic effects of heat strain [2]. In particular, the International Labour Organization estimated that more than 1 billion workers are exposed to high heat episodes, even though not all of these occur during the

**Data Availability Statement:** The minimal anonymized data set necessary to replicate our study is available at Zenodo (DOI: 10.5281/zenodo.10280250).

**Funding:** This work was supported by JSPS KAKENHI(https://www.jsps.go.jp/english/index.html) grant number 21K10449(TK). The funders had no role in study design, data collection and analysis, decision to publish, or preparation of the manuscript.

**Competing interests:** The authors have declared that no competing interests exist.

summer months [3]. The World Health Organization also asserts that "any decline in a worker's performance of daily activities due to heat, cold, or extreme weather should be considered a "health effect" of climate conditions" [4, 5].

In Japan, HRI is a common occupational health problem. Therefore, during the 13th Occupational Safety and Health Program, the Japanese Ministry of Health, Labour and Welfare stated that "the prevention of occupational heat stress is an important goal to reduce the incidences of exertional heat illness during work" [6]. In addition, the ministry declares annual campaigns with the agenda to "Combat Heatstroke at work" to promote efficient practices against HRI; however, despite this intervention, the number of victims of HRI at work has not declined over the past 10 years [7].

Occupational health management comprises three principal factors, namely: work environment control, work practice management, and health care [8]. To minimize occupational HRI, several measures are encouraged, such as on-site temperature measurement, a cooling space set up for work environment control, continuous work time control, availability of adequate equipment such as sunshade, water, and salt intake for work practice management, adequate sleep, breakfast consumption, and disease control for health care [9–11]. However, adequate rehydration appears to be challenging in occupational settings, especially when workers are exposed to extreme heat during work and leisure time (including overnight). This has been indicated by the high prevalence of low hydration status at the onset of work in occupations with high heat stress and the accumulative negative effects on work productivity following consecutive heat stress [12, 13].

Japanese firefighters are engaged in a 24-h rotation duty, which involves various risks, and thus, imposes a high psycho-physiological workload on them [14]. To minimize these risks, firefighters are obliged to wear fireproof garments for their safety [15]. However, even though this clothing provides excellent protection from fire hazards, it also leads to excessive heat accumulation during physical exertion owing to its high thermal resistance and poor vapor permeability [16, 17]. Consequently, many studies have suggested that an evaporative resistance in conditions of high heat load during firefighting may substantially increase their susceptibility to HRI [16, 17]. Therefore, preventive measures are required to reduce the risks of HRI among firefighters working in harsh and hot environments.

Salt intake has been reported as an important measure for work practice management; however, some studies have confused it with dietary salt intake, and hence, considered it a part of health care. Nonetheless, it is worth noting that the salt intake in HRI aims to prevent heat cramps, as shown in an earlier study, wherein we reported that salt intake reduces the risk of HRI [10]. Although younger workers tend to perform fewer actions to prevent HRI [18], an oral rehydration solution during exercise may reduce their susceptibility to HRI [19]. Ideally, salt is consumed with water as an isotonic fluid. However, in Japan, as a result of emphasizing only on salt intake and not supplementation of lost salt, there are cases of overdose of salt in daily life and salt intake before work. In addition, adequate intake of electrolytes and water balance is another matter worth studying [20]. Therefore, this study aimed to investigate the optimal timing of salt intake to prevent HRI during hot outdoor work.

## Materials and methods

### Study design and participants

This observational study included firefighters working at a fire department in Fukushima Prefecture, Japan. The fire department is located coast side of Fukushima Prefecture, and the Fukushima Daiichi-Nuclear Power Plant is located within the jurisdiction of this fire station. There are many areas under the jurisdiction of fire departments with high radiation doses, and

there are many operations in which protective clothing is worn; thus, implementation of measures against heatstroke in the summer is required. A total of 120 people worked in this fire department, and we recruited firefighters according to the following inclusion criteria: those who worked 24 h daily, those aged ≥ 20 years, and those with no past medical history of hypertension and diabetes mellitus that increase the risk of HRI. In total, 58 firefighters were recruited. The included participants were those who participated in outdoor training in the summer and had no past medical history, among whom 28 men agreed to participate in the study. All participants were considered acclimatized to the heat because the recruited subjects were engaged in sweaty exercise at least three times a week. This study was performed in accordance with the "Ethical Guidelines for Medical and Health Research Involving Human Subjects." It was approved by the Ethics Committee of Fukushima Medical University (approval number- 2021–110). Written informed consent was obtained from all participants.

## Data collection

The study period was from July 1 to August 30, 2021, which was announced as "a period to strengthen the prevention of occupational injury" in Japan. The participants completed a multiple-choice questionnaire before and after outdoor training. Before training, the administered questionnaire consisted of questions on effective sleep, alcohol consumption, and body weight. After training, the administered questionnaire included questions on salt intake with options on before, during, and after training or none, water intake, body weight, and subjective HRI symptoms (multiple-choice). Subjective HRI symptoms included muscle pain in the arms or legs, muscle spasms in the arms or legs, intense thirst, decreased urine output, headache, dizziness, lightheadedness, discomfort, weakness, and loss of consciousness. These symptoms were selected based on past published studies [21–23]. Body weight was measured using the Body Composition Analyzer BS-230 (Dretech Co., Ltd., Saitama, Japan). The presence of HRI was determined based on subjective and/or objective symptoms. A weight loss ≥ 1.5% after training was selected as an objective HRI symptom [9]. Water was readily available for the participants to drink freely. We also prepared a commercially available salt tablet (0.108 g salt) at the training site. The participants were allowed to take it at any time. The quantities of water and salt consumed were recorded. Finally, data for a total of 250 working days were obtained.

Participants' age, height, weight, and medical history data were collected through health checkups. Body mass index (BMI) was calculated from the height and weight of the participants. However, medical records for hypertension and diabetes were not documented for the participants of this study.

To account for alcohol consumption, we provided the participants with a brief self-administered diet history questionnaire (BDHQ) in the beginning of July [24]. After they answered the BDHQ, the dietitians confirmed the contents of the questionnaire and interviewed the missing answers directly. Afterward, we calculated alcohol consumption as pure ethanol intake per day.

Details on weather conditions were collected from the Japan Meteorological Agency [25]. In addition, we collected information on dry temperature, relative humidity, and wet-bulb globe temperature (WBGT) every hour at the site. We also collected the maximum and minimum dry temperature and WBGT on the last day and compared the difference between each index.

## Statistical analysis

We used R 4.0.3 [26] for all statistical analyses. Data are presented as mean ± standard deviation (SD) for continuous variables with assumed normal distribution, median (interquartile

**Table 1. Characteristics of the study participants (N = 28).**

| Items | Data |
|---|---|
| Age, year, mean ± SD[a] | 31.0 ± 7.7 |
| Body mass index, kg/m², mean ± SD[a] | 25.0 ± 3.1 |
| Number of working days, days, median (IQR)[b] | 7 (4–12) |
| Training time, h, mean ± SD[b] | 1.64 ± 0.64 |
| Sleep duration, hours, mean ± SD[b] | 6.12 ± 0.99 |
| Water intake, time, median (IQR)[b] | 350 (0–500) |
| Daily salt intake, g/day, median (IQR)[c] | 12.6 (10.2–14.7) |
| Alcohol intake, g/day, median (IQR)[c] | 12.8 (2.9–28.9) |

SD, standard deviation; IQR, interquartile range.

a: Data were obtained from medical records

b: Data were obtained from questionnaires administered after training

c: Data were obtained from a brief self-administered diet history questionnaire

range [IQR]) for continuous variables without assumed normal distribution, or n (%) for categorical variables.

From the results of the BDHQ, we calculated the daily alcohol intake for 4 days/week because Japanese firefighters are engaged in 24 h shifts and can drink alcohol on 4 days/week [27].

We classified the salt intake pattern into four groups: no intake, before training, before and during training, and during training. A logistic mixed effects model [28, 29] was used to model repeated measurements over determine factors related to HRI. Adequate sleep, alcohol consumption, and salt intake pattern were selected as the fixed effects. Participants ID, area, and daily maximum WBGT (over 28°C) were selected as the random effects. Odds ratios (ORs) and their 95% confidence intervals (95% CI) were calculated. We used "lme4" packages for this model.

## Results

The characteristics of the study participants are shown in Table 1. The mean age of the participants was 31.0 ± 7.7 years, and their mean BMI was 25.0 ± 3.1 kg/m². The median working duration was 7 days (4–12 days); the mean training time was 1.64 ± 0.64 h/day; the median salt intake was 12.6 (10.2–14.7) g/day; and the median alcohol intake was 12.8 (2.9–28.9) g/day. Twenty-six participants (92.9%) responded that they consumed alcohol, and after adjusting for the number of current drinkers in the group, the median alcohol intake was 25.6 (6.6–50.7) g/day. Of the 26 drinkers, the median intake for 10 participants was over 40 g/day, which was defined as excess alcohol intake [30]. Notably, there was no significant relationship between daily dietary salt intake and HRI prevalence.

The environmental data for each training site are listed in Table 2. The median temperature and WGBT were recorded as 26.9°C (24.4–29.5) and 26.3°C (23.8–28.7) at Site 1, 27.2°C (24.7–31.2), and 26.2°C (22.8–30.1) at Site 2, and 27.1°C (24.8–29.3) and 26.4°C (24.0–28.2) at Site 3, respectively. No significant differences were observed among the sites for temperature and WGBT (p = 0.658 and 0.953, respectively).

The study was performed within 250 working days, and we detected 28 HRI symptoms (11.2%) during the study period. The symptom frequency was calculated as a percentage of the days of symptom occurrence against the total number of days of the study. The most frequent symptom was weight loss ≥ 1.5% (n = 10, 4.0%). Other frequent symptoms included intense

**Table 2. Environmental status of each site.**

| | Site 1 (n = 72) | Site 2 (n = 21) | Site 3 (n = 24) | p-value[a] |
|---|---|---|---|---|
| Maximum temperature (°C) | 26.9 (24.4–29.5) | 27.2 (24.7–31.2) | 27.1 (24.8–29.3) | 0.658 |
| Maximum WBGT (°C) | 26.3 (23.8–28.7) | 26.2 (22.8–30.1) | 26.4 (24.0–28.2) | 0.953 |

WBGT, wet-bulb globe temperature.

All data are presented as median (interquartile range).

[a] p-values were calculated using the Kruskal–Wallis test

**Table 3. Proportions of heat-related illness symptoms.**

| Symptoms | Total | Site 1 | Site 2 | Site 3 |
|---|---|---|---|---|
| | N (%) | n (%) | n (%) | n (%) |
| | N = 250 | n = 132 | n = 27 | n = 90 |
| *Objective* | | | | |
| Weight loss ≥ 1.5% | 10 (4.0) | 3 (2.3) | 0 (0.0) | 7 (7.7) |
| *Subjective* | | | | |
| Intense thirst | 9 (3.6) | 4 (3.0) | 0 (0.0) | 5 (5.5) |
| Dizziness | 8 (3.2) | 4 (3.0) | 1 (3.7) | 3 (3.3) |
| Discomfort | 7 (2.8) | 3 (2.3) | 0 (0.0) | 4 (4.4) |
| Headache | 5 (2.0) | 1 (0.8) | 0 (0.0) | 4 (3.3) |
| Pains in the arms or legs | 2 (0.8) | 0 (0.0) | 0 (0.0) | 2 (2.2) |
| Decreased urine output | 1 (0.4) | 0 (0.0) | 0 (0.0) | 1 (1.1) |
| Lightheadedness | 1 (0.4) | 0 (0.0) | 0 (0.0) | 1 (1.1) |
| Weakness | 1 (0.4) | 0 (0.0) | 0 (0.0) | 1 (1.1) |
| Loss of consciousness | 1 (0.4) | 0 (0.0) | 0 (0.0) | 1 (1.1) |
| Muscle spasms in the arms or legs | 0 (0.0) | 0 (0.0) | 0 (0.0) | 0 (0.0) |

thirst (n = 9, 3.6%), dizziness (n = 8, 3.2%), discomfort (n = 7, 2.8%), and headache (n = 5, 2.0%) (Table 3).

Logistic mixed effect model analysis revealed that salt intake before training (OR: 5.893, 95% CI: 1.407–24.675), and salt intake before and during training (OR: 22.889, 95% CI: 4.276–122.516) were positively associated with HRI symptoms (Table 4).

**Table 4. Logistic mixed effect model of heat-related illness risk.**

| Item (Fixed effect) | Adjusted OR (95% CI) |
|---|---|
| Adequate sleep | -0.997 (0.288–3.450) |
| Alcohol drinking | 2.133 (0.855–5.320) |
| Water intake | 1.273 (0.429–3.782) |
| Salt intake (reference: none) | |
| Before training | 5.893 (1.407–24.675) |
| Before and during training | 22.889 (4.276–0122.516) |
| During training | 2.562 (0.539–12.185) |

OR, odds ratio; CI, confidence interval; BMI, body mass index; WBGT, wet-bulb globe temperature.

## Discussion

In this study, we investigated the risk factors for HRI among firefighters during summer training and the effect of the timing of salt intake on the risks of HRI. We found that salt intake before training positively affects HRI, suggesting that preventive salt intake may lead to HRI. Even though the importance of salt intake in preventing HRI is well known, the specific method of ingestion has not been sufficiently studied [10, 19, 20, 31]. Therefore, to the best of our knowledge, this is the first study to focus on the relationship between the timing of salt intake and HRI.

Firefighters are engaged in a 24 h work schedule, and training for emergency services is one of their important daily tasks [15, 32]. However, this training imposes a heavy load on them, especially in a harsh environment, thereby validating the observation of a high HRI occurrence in this study. Besides, this present study under these conditions also reveals that salt intake may be a potent determinant of HRI risks.

Increased plasma osmolality indicates dehydration, and this may be corrected by sufficient water intake to restore any deficit in the intracellular fluid levels. A previous study demonstrated that after exercise, the ingestion of carbohydrate-electrolyte solutions improved the plasma volume, serum osmolality, and serum $Na^+$ concentration while reducing water loss through urination more efficiently than those following the intake of only water [33]. In addition, another study also reported that serum sodium, chloride, and potassium levels showed no remarkable difference after consuming different types of beverages (diet cola, carbohydrate-electrolyte solution, and water) [34]. Consequently, these studies mentioned above indicate that the electrolyte concentration in the intracellular fluid is strictly controlled by homeostasis and that the electrolyte deficit in the body mainly results from the loss of electrolytes from the extracellular fluid. Nonetheless, the repercussions of excessive salt intake in terms of electrolyte restoration are still under discussion. In particular, the relationship between electrolyte loss due to sweating and the benefits of extracellular electrolyte administration is currently being researched.

Salt intake has been encouraged to prevent HRI [7, 10, 19]. This was validated in a prior study that reported that in the absence of sodium intake during training, the muscle cramp threshold frequency decreased significantly, thus, indicating an increase in the susceptibility to cramps [19]. However, both sodium and water are necessary to replenish deficits in the extracellular fluid to prevent dehydration. Notably, a study on fluid-deprived Zucker rats predisposed to hypertension showed that at a dehydration level of > 3%, the rats initially chose hypotonic sodium chloride (NaCl) solution before consuming the almost isotonic NaCl solution to replenish their lost body fluids. Whereas the other group of animals exposed to hypertonic NaCl solution at the same time justified the absence of salt intake induction by a significant increase in water consumption, most probably to maintain constant body fluid balance [31]. Conversely, excess dietary sodium may actually decrease plasma volume, even with a controlled fluid intake [20]. When excessive salt is ingested, the intracellular fluid moves to the extracellular fluid compartment to lower the salt concentration in the blood, thereby inducing urination to mediate the salt excretion [35]. Furthermore, increased urine sodium levels and urine volume as a result of high dietary sodium intake may also cause decreased plasma renin and aldosterone [36]. Overall, high dietary sodium only increases urine volume and is not recommended for preventing HRI. Therefore, the National Institute for Occupational Safety and Health declared that supplemental dietary sodium must be used judiciously to prevent further dehydration and electrolyte depletion in workers [20].

Generally, alcohol intake is strongly associated with HRI [20]. Ethanol inhibits anti-diuretic hormones, and increased alcohol consumption has been shown to increase gastrointestinal

permeability, thus, enhancing the risks of dehydration [20, 37]. Moreover, alcohol also induces reactive hypoglycemia by exacerbating insulin secretion in the presence of a high carbohydrate meal [38]. Therefore, measures to create awareness of the risks associated with alcohol consumption and provide guidance to limit drinking on the day before work are necessary. Our results did not find a significant relationship between alcohol intake and HRI. The estimated daily alcohol intake among the study subjects was 25.6 g/day, with 10 subjects estimated to consume over 40 g/day. Considering the high alcohol consumption among the study participants, a significant relationship may not have been observed in this study.

In this study, weight loss was selected as an indicator of fluid loss. Heavy sweating is often a symptom of fluid loss [22, 23], as heat acclimation adjusts the body's sweating mechanism (such as the threshold body temperature for sweating) [39]. Notably, because the amount of abnormal sweating varies greatly among individuals, the subjective index was judged to be unreliable. In addition, the amount of sweat produced depends on the state of hydration, and progressive dehydration results in lower sweat production [20]. Nevertheless, a previous study stated that dehydration over 1.5% could reduce performance, even during short-duration exercise [40]. Therefore, weight loss can be considered an indicator of dehydration.

This study has some limitations. First, we did not consider each individual's workload. The risks of HRI should be considered based on environmental conditions and workloads [41]. Detected HRI symptoms differed between each site. We considered that all participants engaged in a heavy workload during training from additional hearing; however, we could not estimate the actual workload in each study site. Therefore, the estimation of workload as an objective method should be further studied. Second, this study targeted only firefighters. As previously mentioned, firefighters engage in special working shifts and are exposed to heavy workloads on a daily basis. Therefore, a simple generalization of this result requires caution; however, results obtained in a population that is adapted to such harsh environments are likely to provide suggestions for other populations as well. Third, the participants in the study belong to one fire department. Target populations were not large enough so, we conducted repeated measurements. Conversely, the current result should be treated with caution considering the volume of the data generated in the study.

## Conclusion

Our results indicate that inappropriate timing of salt intake may increase the risks of HRI. Thus, timely intake of salt in adequate amounts may prevent these risks. Adequate timing and amount of salt intake should be investigated in future studies.

## Acknowledgments

We would like to thank Editage (www.editage.com) for English language editing.

## Author Contributions

**Conceptualization:** Takeyasu Kakamu, Teruna Ito.

**Data curation:** Takeyasu Kakamu.

**Formal analysis:** Takeyasu Kakamu.

**Funding acquisition:** Takeyasu Kakamu.

**Investigation:** Takeyasu Kakamu, Teruna Ito, Shota Endo.

**Methodology:** Takeyasu Kakamu, Shota Endo, Tomoo Hidaka.

**Project administration:** Takeyasu Kakamu.

**Resources:** Takeyasu Kakamu, Teruna Ito.

**Software:** Takeyasu Kakamu.

**Supervision:** Tetsuhito Fukushima.

**Validation:** Tomoo Hidaka, Yusuke Masuishi, Hideaki Kasuga.

**Visualization:** Takeyasu Kakamu.

**Writing – original draft:** Takeyasu Kakamu.

**Writing – review & editing:** Teruna Ito, Shota Endo, Tomoo Hidaka, Yusuke Masuishi, Hideaki Kasuga, Tetsuhito Fukushima.

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
