## [Decision Letter · Decision Letter 0]

28 Mar 2023

PONE-D-23-02599Inappropriate timing of salt intake increases the risk of heat-related illness: An observational studyPLOS ONE

Dear Dr. Kakamu,

Thank you for submitting your manuscript to PLOS ONE. After careful consideration, we feel that it has merit but does not fully meet PLOS ONE’s publication criteria as it currently stands. Therefore, we invite you to submit a revised version of the manuscript that addresses the points raised during the review process.

We look forward to receiving your revised manuscript.

Kind regards,

William M. Adams

Academic Editor

PLOS ONE

“We would like to thank Editage (www.editage.com) for English language editing.

This work was supported by JSPS KAKENHI (grant number 21K10449).”

“This work was supported by JSPS KAKENHI(https://www.jsps.go.jp/english/index.html) grant number 21K10449　(TK). The funders had no role in study design, data collection and analysis, decision to publish, or preparation of the manuscript.”

Reviewers' comments:

Reviewer's Responses to Questions

**Comments to the Author**

1. Is the manuscript technically sound, and do the data support the conclusions?

Reviewer #1: Partly

Reviewer #2: Yes

2. Has the statistical analysis been performed appropriately and rigorously? 

Reviewer #1: No

Reviewer #2: No

3. Have the authors made all data underlying the findings in their manuscript fully available?

Reviewer #1: No

Reviewer #2: No

4. Is the manuscript presented in an intelligible fashion and written in standard English?

Reviewer #1: Yes

Reviewer #2: Yes

5. Review Comments to the Author

Reviewer #1: Thank you very much for the opportunity to review this very interesting paper.

However, I believe that two main points need to be fundamentally modified

(1) Sample size

The main analysis is a multivariate logistic analysis with 7 covariates, but you have not considered whether the sample size of 28 participants is sufficient for statistical analysis.

I apologize if the number of participants is incorrect. However, even in that case, it is difficult to understand the attributes of the participants, and we would appreciate clarification.

2) Attributes of each group in the Salt intake

Since the details of each of the groups are not clear, it is not possible to determine whether Salt intake can be considered a significant factor.

I am sure there are other points to consider, but if you are going to submit a revised submission, I would appreciate it if you could respond to the above two points as soon as possible and then review it again.

Reviewer #2: 1. Specific errors here:

37 line: indiacte

73 line: minimze

84-85 line: salt intake in HRI aims

89 line: emphasizing

102: in accordance with

239: may actually

271: a daily basis

2. Material and Methods /

2.1. Study design and participants.

Please include inclusion criteria, heat acclimation criteria. Describe the selected Fukushima Prefectural Fire Department sites, total number of possible participants. Add in the annexes the Informed Consent and the Approval of the Ethics Committee.

2.3 Data Collection.

Please add in the annexes the questionnaires before and after the training, also clarify how they took the value during the training.

Define the variables contained between lines 111 to 114. Define the parameters under which HRI was defined. How the salt tablets were prepared and how they were administered.

Define how weight and height measurements were taken, what equipment was used.

What information was taken from the medical history.

Add the BDHQ questionnaire, which method was used to calculate alcohol consumption, number of responses not given.

Were the dietitians previously trained?

Were the weather conditions evaluated with those taken at the site and their relationship with HRI?

3. Results.

How did you calculate the concentration of salt in the diet and if you took salt tablets, how and at what time?

It would be possible to add more sociodemographic information by site, including gender, to Table 1.

What time of day did the training take place?

In table 3, can the information be disaggregated by each site?

How was the information collected during the training?

4. Conclusion

It is suggested to support the conclusions with part of the information placed in the summary.

5. Additional comments for the author.

Personally, I consider it to be excellent research whose results will help improve the treatment and prevention of HRI.

I believe that the research has much more data to contribute, which were taken during the field work, it is possible that its analysis and a more robust correlation of the variables could increase the conclusions, achieving greater indications for the prevention of this pathology. .

6. PLOS authors have the option to publish the peer review history of their article (what does this mean?). If published, this will include your full peer review and any attached files.

Reviewer #1: No

Reviewer #2: No

---

## [Author Response · Author response to Decision Letter 0]

25 Apr 2023

Response to reviewer

Dear Reviewers,

We appreciate for your constructive comments on our paper. We have carefully revised the manuscript text based on your comments. The point to point responses are below.

Reviewer #1: Thank you very much for the opportunity to review this very interesting paper.

However, I believe that two main points need to be fundamentally modified

(1) Sample size

The main analysis is a multivariate logistic analysis with 7 covariates, but you have not considered whether the sample size of 28 participants is sufficient for statistical analysis.

I apologize if the number of participants is incorrect. However, even in that case, it is difficult to understand the attributes of the participants, and we would appreciate clarification.

Response: To clarify, there were 28 study participants. In total, they recorded outdoor training for 250 days. The sample size sufficient for statistical analysis was 250 working days. In the Methods section of the revised manuscript, we have mentioned the total sample size.

2) Attributes of each group in the Salt intake

Since the details of each of the groups are not clear, it is not possible to determine whether Salt intake can be considered a significant factor.

Response: We apologize for this lack of explanation. We have added this detail in the revised manuscript.

I am sure there are other points to consider, but if you are going to submit a revised submission, I would appreciate it if you could respond to the above two points as soon as possible and then review it again.

Response: Thank you for your comments. We have carefully reviewed your comments and addressed them in the revised manuscript.

Reviewer #2: 1. Specific errors here:

37 line: indiacte

73 line: minimze

84-85 line: salt intake in HRI aims

89 line: emphasizing

102: in accordance with

239: may actually

271: a daily basis

Response: We apologize for these typographical errors. We have corrected them in the revised manuscript. In addition, we have had the manuscript checked by a native English speaker from a professional editing company to ensure that there are no remaining grammatical or syntax errors. We hope that the revised manuscript is now suitable for publication.

2. Material and Methods /

2.1. Study design and participants.

Please include inclusion criteria, heat acclimation criteria. Describe the selected Fukushima Prefectural Fire Department sites, total number of possible participants. Add in the annexes the Informed Consent and the Approval of the Ethics Committee.

Response: In the revised manuscript, we have detailed the inclusion criteria, heat acclimation criteria, selected fire department, total number of fire departments, and recruited subjects. We have also submitted the annexes you have mentioned to the editorial office.

2.3 Data Collection.

Please add in the annexes the questionnaires before and after the training, also clarify how they took the value during the training.

Response: The original version of the questionnaire is written in Japanese. In this resubmission, we have submitted the questionnaire as an annexed file. Participants administered to questionnaire after the training.

Define the variables contained between lines 111 to 114. Define the parameters under which HRI was defined. How the salt tablets were prepared and how they were administered.

Response: As per your comment, we have clarified the definition of HRI in the Statistical analysis section . Details regarding the salt tablets are described in line 131.

Define how weight and height measurements were taken, what equipment was used.

Response: Data on height were obtained from health checkup records. Body weight was measured using Body Composition Analyzer BS-230 (Dretech Co., Ltd, Saitama, Japan). We added these details in the revised manuscript (lines 127–128).

What information was taken from the medical history.

Response: From the medical history, we collected information on the history of hypertension and diabetes mellitus because these diseases increase the risk of HRI.

Add the BDHQ questionnaire, which method was used to calculate alcohol consumption, number of responses not given.

Response: We cannot provide the BDHQ as it is a copyrighted material. We have instead included a reference paper demonstrating the reliability of the BDHQ in nutritional surveys.

Were the dietitians previously trained?

Response: Yes, one dietitian (co-author TI) was trained to evaluate the BDHQ.

Were the weather conditions evaluated with those taken at the site and their relationship with HRI?

Response: Yes, daily maximum WBGT was included as a confounding factor, and it indicated a negative result.

3. Results.

How did you calculate the concentration of salt in the diet and if you took salt tablets, how and at what time?

Response: The concentration of salt in the diet was estimated from the BDHQ. The BDHQ was administered in the beginning of July.

It would be possible to add more sociodemographic information by site, including gender, to Table 1.

Response: All firefighters in the study department were men. Since one participant trained at multiple sites, the demographic data for each site could not be described.

What time of day did the training take place?

Response: The fire department scheduled training time twice daily (morning and afternoon), and all firefighter engaged training once in a day without emergency dispatch.

In table 3, can the information be disaggregated by each site?

Response: We have added more information in Table 3 and corrected a typographical error. However, this error did not affect the results of logistic regression analysis.

How was the information collected during the training?

Response: As described in lines 118-119, study participants answered a self-administered questionnaire before and after training.

4. Conclusion

It is suggested to support the conclusions with part of the information placed in the summary.

Response: Yes, we have revised the conclusions section and unified the content with that in the Abstract.

5. Additional comments for the author.

Personally, I consider it to be excellent research whose results will help improve the treatment and prevention of HRI.

I believe that the research has much more data to contribute, which were taken during the field work, it is possible that its analysis and a more robust correlation of the variables could increase the conclusions, achieving greater indications for the prevention of this pathology.

Response: Thank you for your favorable review of our manuscript. We have carefully revised our paper to sufficiently address all your comments.

---

## [Decision Letter · Decision Letter 1]

6 Jun 2023

PONE-D-23-02599R1Inappropriate timing of salt intake increases the risk of heat-related illness: An observational studyPLOS ONE

Dear Dr. Kakamu,

Thank you for submitting your manuscript to PLOS ONE. After careful consideration, we feel that it has merit but does not fully meet PLOS ONE’s publication criteria as it currently stands. Therefore, we invite you to submit a revised version of the manuscript that addresses the points raised during the review process.

ACADEMIC EDITOR: Please address the comments and concerns posed by Reviewer #1

We look forward to receiving your revised manuscript.

Kind regards,

William M. Adams

Academic Editor

PLOS ONE

Journal Requirements:

Additional Editor Comments:

Please address the comments and concerns posed by reviewer #1.

Reviewers' comments:

Reviewer's Responses to Questions

**Comments to the Author**

1. If the authors have adequately addressed your comments raised in a previous round of review and you feel that this manuscript is now acceptable for publication, you may indicate that here to bypass the “Comments to the Author” section, enter your conflict of interest statement in the “Confidential to Editor” section, and submit your "Accept" recommendation.

Reviewer #1: All comments have been addressed

Reviewer #2: All comments have been addressed

2. Is the manuscript technically sound, and do the data support the conclusions?

Reviewer #1: No

Reviewer #2: Yes

3. Has the statistical analysis been performed appropriately and rigorously? 

Reviewer #1: No

Reviewer #2: Yes

4. Have the authors made all data underlying the findings in their manuscript fully available?

Reviewer #1: Yes

Reviewer #2: Yes

5. Is the manuscript presented in an intelligible fashion and written in standard English?

Reviewer #1: No

Reviewer #2: Yes

6. Review Comments to the Author

Reviewer #1: The description of the statistical analysis is still confusing.

Regarding the sample size, which we pointed out in the last issue, the process of calculating the required number of 250 cases is not clear.

However, if I were to accept your assertion, then 28 participants would require 250 cases, which would mean multiple data collections from a single patient.

From the statement "Table 4. logistic regression analysis of heat-related illness risk.", I assume that a multivariate logistic analysis was performed.

However, in principle, in a multivariate logistic analysis, the outcome is measured once per patient.

From your description, I can only follow the above inference. From this inference, I cannot say that you have performed the statistical analysis correctly, and I have no choice but to reject your paper.

Reviewer #2: All my comments, questions, concerns and recommendations were fully resolved. It is an excellent article. I appreciate the opportunity given for its review and congratulations to the authors.

7. PLOS authors have the option to publish the peer review history of their article (what does this mean?). If published, this will include your full peer review and any attached files.

Reviewer #1: No

Reviewer #2: No

---

## [Author Response · Author response to Decision Letter 1]

20 Aug 2023

We have revised the description related to Table 4 following the results of the recalculations.

---

## [Decision Letter · Decision Letter 2]

6 Nov 2023

PONE-D-23-02599R2Inappropriate timing of salt intake increases the risk of heat-related illness: An observational studyPLOS ONE

Dear Dr. Kakamu,

Thank you for submitting your manuscript to PLOS ONE. After careful consideration, we feel that it has merit but does not fully meet PLOS ONE’s publication criteria as it currently stands. Therefore, we invite you to submit a revised version of the manuscript that addresses the points raised during the review process.

ACADEMIC EDITOR:Dear Authors,Kindly address the concern of Reviewer 1 on your submission. Specifically, you need to explain why the switch of the regression methods of statistical analysis. I would strongly suggest that you give:1. Some supporting citation(s) for the new statistical approach adopted.2. Limitations of the study, clearly indicating that the results reported should be treated with caution considering the volume of the data generated in the study.3. Directions for future research under CONCLUSIONS.  Please submit your revised manuscript by Dec 21 2023 11:59PM. If you will need more time than this to complete your revisions, please reply to this message or contact the journal office at plosone@plos.org. Please include the following items when submitting your revised manuscript:A rebuttal letter that responds to each point raised by the academic editor and reviewer(s). You should upload this letter as a separate file labeled 'Response to Reviewers'.A marked-up copy of your manuscript that highlights changes made to the original version. You should upload this as a separate file labeled 'Revised Manuscript with Track Changes'.An unmarked version of your revised paper without tracked changes. You should upload this as a separate file labeled 'Manuscript'.If applicable, we recommend that you deposit your laboratory protocols in protocols.io to enhance the reproducibility of your results. Protocols.io assigns your protocol its own identifier (DOI) so that it can be cited independently in the future. For instructions see: https://journals.plos.org/plosone/s/submission-guidelines#loc-laboratory-protocols. Additionally, PLOS ONE offers an option for publishing peer-reviewed Lab Protocol articles, which describe protocols hosted on protocols.io. Read more information on sharing protocols at https://plos.org/protocols?utm_medium=editorial-email&utm_source=authorletters&utm_campaign=protocols.

We look forward to receiving your revised manuscript.

Kind regards,

Timothy Omara, PhD

Academic Editor

PLOS ONE

Journal Requirements:

Reviewers' comments:

Reviewer's Responses to Questions

**Comments to the Author**

1. If the authors have adequately addressed your comments raised in a previous round of review and you feel that this manuscript is now acceptable for publication, you may indicate that here to bypass the “Comments to the Author” section, enter your conflict of interest statement in the “Confidential to Editor” section, and submit your "Accept" recommendation.

Reviewer #1: All comments have been addressed

2. Is the manuscript technically sound, and do the data support the conclusions?

Reviewer #1: Partly

3. Has the statistical analysis been performed appropriately and rigorously? 

Reviewer #1: I Don't Know

4. Have the authors made all data underlying the findings in their manuscript fully available?

Reviewer #1: Yes

5. Is the manuscript presented in an intelligible fashion and written in standard English?

Reviewer #1: No

6. Review Comments to the Author

Reviewer #1: I addressed the questions regarding logistic analysis and study design in the previous peer review.

I had expected that you would revise the study design, but you have utilized a different statistical technique.

There was no response to my comments on what led to the use of this statistical technique.

Since I am not a statistician, I cannot judge the validity of the conclusions you have drawn from the results of this analysis without comment.

I have assigned a Major Revision for this peer review.

If the manuscript is to be resubmitted, it needs to be evaluated by reviewers other than me.

7. PLOS authors have the option to publish the peer review history of their article (what does this mean?). If published, this will include your full peer review and any attached files.

Reviewer #1: No

---

## [Author Response · Author response to Decision Letter 2]

11 Dec 2023

Response to the reviewer

Dear Editor,

We appreciate your constructive comments on our paper. We have carefully revised the manuscript text based on your comments. We explained why we selected mixed-effect model in the response for reviewer #1.

1. Some supporting citation(s) for the new statistical approach adopted.

Response: Thank you for your suggestion. We added references that support our new statistical method (citation No28 and 29).

2. Limitations of the study, clearly indicating that the results reported should be treated with caution considering the volume of the data generated in the study.

Response: We have noted the need for caution regarding the handling of the results in the limitation section. 

3. Directions for future research under CONCLUSIONS.

Response: We added sentence in conclusions section.

Dear Reviewers,

We appreciate your constructive comments on our paper. We have carefully revised the manuscript text based on your comments. Our point-to-point responses are below.

Reviewer #1: I addressed the questions regarding logistic analysis and study design in the previous peer review.

I had expected that you would revise the study design, but you have utilized a different statistical technique.

There was no response to my comments on what led to the use of this statistical technique.

Since I am not a statistician, I cannot judge the validity of the conclusions you have drawn from the results of this analysis without comment.

I have assigned a Major Revision for this peer review.

If the manuscript is to be resubmitted, it needs to be evaluated by reviewers other than me.

Response: The mixed-effects model is a statistical method used in studies where the same individual is measured repeatedly, such as in the current study. Information that can identify individuals and information that takes the same value for multiple subjects are referred to as fixed effects, while information that varies by measurement is treated as random effects and incorporated as variables. We have added a citation for a paper that utilizes these mixed-effects models.

---

## [Editor Report · Decision Letter 3]

13 Dec 2023

Inappropriate timing of salt intake increases the risk of heat-related illness: An observational study

PONE-D-23-02599R3

Dear Dr. Kakamu,

We’re pleased to inform you that your manuscript has been judged scientifically suitable for publication and will be formally accepted for publication once it meets all outstanding technical requirements.

Kind regards,

Timothy Omara, PhD

Academic Editor

PLOS ONE
---

## [Editor Report · Acceptance letter]

20 Dec 2023

PONE-D-23-02599R3 

PLOS ONE

Dear Dr. Kakamu, 

I'm pleased to inform you that your manuscript has been deemed suitable for publication in PLOS ONE. Congratulations! Your manuscript is now being handed over to our production team.

Kind regards, 

on behalf of

Dr. Timothy Omara 

Academic Editor

PLOS ONE